# Synergy of Regenerative Periodontal Surgery and Orthodontics Improves Quality of Life of Patients with Stage IV Periodontitis: 24-Month Outcomes of a Multicenter RCT

**DOI:** 10.3390/bioengineering10060695

**Published:** 2023-06-07

**Authors:** Karin Jepsen, Christina Tietmann, Conchita Martin, Eric Kutschera, Andreas Jäger, Peter Wüllenweber, Lorena Gaveglio, Daniele Cardaropoli, Ignacio Sanz-Sánchez, Rolf Fimmers, Søren Jepsen

**Affiliations:** 1Department of Periodontology, Operative and Preventive Dentistry, University of Bonn, Welschnonnenstrasse 17, 53111 Bonn, Germany; tietmann@paro-aachen.de (C.T.);; 2Private Practice for Periodontology, Krefelder Strasse 73, 52070 Aachen, Germany; 3BIOCRAN Research Group, University Complutense of Madrid, 28040 Madrid, Spain; 4Department of Orthodontics, University of Bonn, Welschnonnenstrasse 17, 53111 Bonn, Germany; 5Private Practice for Orthodontics, Theaterstraße 98-102, 52062 Aachen, Germany; 6Private Practice, Corso Galileo Ferraris 148, 10129 Turino, Italy; 7ETEP Research Group, University Complutense of Madrid, 28040 Madrid, Spain; 8Institute for Medical Biometry, Informatics and Epidemiology, University of Bonn, 53127 Bonn, Germany

**Keywords:** oral health-related quality of life, oral rehabilitation, regenerative periodontal therapy, orthodontic tooth movement, pathologic tooth migration, bovine bone mineral, collagen, enamel matrix derivative, stage IV periodontitis, randomized clinical trial

## Abstract

In stage IV periodontitis patients with pathologic tooth migration (PTM), interdisciplinary treatment includes regenerative periodontal surgery (RPS) with an application of biomaterials and orthodontic therapy (OT) to restore function, esthetics and thereby quality of life (QoL). In a 24-month randomized trial we explored the synergy between regenerative medicine and biomechanical force application. The following methods were used: Forty-three patients had been randomized to a combined treatment comprising RPS and subsequent OT starting either 4 weeks (early OT) or 6 months (late OT) post-operatively. Clinical periodontal parameters and oral health-related QoL (GOHAI) were recorded up to 24 months. We obtained the following results: Mean clinical attachment gain (∆CAL ± SD) was significantly higher with early OT (5.96 ± 2.1 mm) versus late OT (4.65 ± 1.76 mm) (*p* = 0.034). Pocket closure (PPD ≤ 4 mm) was obtained in 91% of defects with early OT compared to 90% with late OT. GOHAI-scores decreased significantly from 26.1 ± 7.5 to 9.6 ± 4.7 (early OT) and 25.1 ± 7.1 to 12.7 ± 5.6 (late OT). Inconclusion, teeth severely compromised by intrabony defects and PTM can be treated successfully by RPS followed by early OT with the advantage of an overall reduced treatment time. As a result of the combined periodontal-orthodontic therapy, the oral health-related QoL of patients was significantly improved. Early stimulation of wound healing with orthodontic forces had a favorable impact on the outcomes of regenerative periodontal surgery.

## 1. Introduction

In stage IV periodontitis patients with pathologic tooth migration (PTM), the sole treatment of periodontitis is usually not sufficient to restore oral health, correct masticatory dysfunction/malocclusion and improve their quality of life (QoL). The periodontal status of these patients is characterized by a similar severity and complexity in terms of inflammation, attachment and bone loss, as in stage III periodontitis, but may require a combined periodontal/orthodontic treatment (OT) for oral rehabilitation in order to restore function and esthetics [1,2,3,4].

It is well established that a regenerative periodontal treatment of intrabony defects can be successfully performed using various surgical procedures and biomaterials [5,6,7,8,9], provided that periodontal inflammation is under control by means of steps 1 and 2 of periodontal therapy. The combination of periodontal regenerative surgery (RPS) and consecutive orthodontic tooth movements in stage IV periodontitis patients was found to be efficient effective in the short-term [10], and the outcomes to be stable for up to 10 years under the premise of an adherence to a strict oral hygiene/maintenance protocol [11,12,13,14,15].

The optimal time interval between regenerative periodontal surgery and the initiation of OT has always been a matter of debate. Pini Prato and Chambrone (2020) [16] proposed waiting until the endpoint of regenerative therapy has been reached in order not to interfere with periodontal wound healing. In contrast, other authors have suggested a “stimulating” effect of early orthodontic tooth movement on the regenerative outcomes [17,18,19]. Several case reports [20,21,22,23,24], and a randomized clinical trial (RCT) [10], demonstrate that teeth severely compromised by intrabony defects and PTM can be treated successfully by regenerative surgery followed by early OT, with the advantage of an overall reduced treatment time.

Nevertheless, a gain in clinical attachment (CAL) and radiographic bone level, as well as reduction in probing pocket depths (PPD) and bleeding on probing (BOP) alone, may not be sufficient to evaluate the overall success of stage IV periodontitis treatment. Patient-related outcomes (PROMs) as “true endpoints” are reported to be equally or more relevant to patients’ daily lives [25,26] and it has been suggested that appropriate oral health-related quality of life (OHrQoL) outcomes should be included in the design of clinical studies [27]. Based on the rationale of a broader view of oral health and its rehabilitation, a number of tools have been introduced in order to measure the extent to which oral conditions affect an individual’s behavior and social life, as well as to complement the conventional clinical assessments of oral health [28,29,30,31,32,33].

Several studies reported that patients with more severe periodontitis rated their OHrQoL as poorer than those who had less severe periodontitis [29,34,35,36]. In addition, a positive perception by patients of the outcomes of long-term supportive therapy after regenerative surgery could be shown [37].

Malocclusion is an important and prevalent oral health problem worldwide [38,39] and has a negative impact on OHrQoL [40,41]. However, at present, there are no data from studies available that have evaluated the impact on OHrQoL in stage IV periodontitis with pathological tooth migration. Likewise, to the best of our knowledge, no studies have investigated the impact on OHrQoL of a combined periodontal and orthodontic treatment to restore function and esthetics in these patients [4].

A recent multi-center randomized trial [10] evaluated the periodontal outcomes of regenerative surgery in stage IV periodontitis patients in combination with staged orthodontic therapy after 12 months. Secondary outcomes of the study protocol also included OHrQoL measures at the baseline and up to 24 months. Here, we report the impact of the combined periodontal–orthodontic treatment on changes of clinical periodontal parameters and oral health-related quality of life.

## 2. Materials & Methods

### 2.1. Study Design and Patients

The present manuscript reports secondary outcomes of a prospective multicenter, multinational, randomized, parallel-group clinical trial (ClinicalTrials.gov, identifier: NCT 02761668) after 24 months. Special emphasis was given to patient-reported outcomes. The study protocol had been previously approved by the respective ethical committees for human subject trials from the centers participating in the study. The lead ethics committee was at the University of Bonn (code 034/16).

In brief, 43 patients with stage IV periodontitis were periodontally treated (steps 1–3 of periodontal therapy). Teeth with pathologic tooth migration and intrabony defects received regenerative periodontal surgery as described by Cortellini and Tonetti [5], followed by orthodontic treatment (OT) initiated 4 weeks after regenerative surgery (early OT, *n* = 23 patients) or 6 months after regenerative surgery (late OT, *n* = 20 patients) (Figure 1).

The details of the study protocol were presented in a previous paper reporting 12-month clinical results [10].

### 2.2. Minimally Invasive Regenerative Periodontal Surgery

Microsurgical approaches, adapted to the treatment algorithm by Cortellini and Tonetti (2015), were applied to access the defects. A bone substitute (DBBMc, Bio Oss^®^ Collagen; Geistlich, Wolhusen, Switzerland) was used to fill the defect and to prevent a soft tissue collapse. In non-contained defects, a collagen membrane (Bio Gide^®^Perio; Geistlich, Wolhusen, Switzerland) was applied. An enamel matrix derivative (EMD, Emdogain^®^; Straumann, Basel, Switzerland) was used for contained defects. Suturing was accomplished with non-resorbable 6-0 and 7-0 monofilament material (e-PTFE, W. L. Gore & Associates, Flagstaff, AZ, USA) by internal offset vertical mattress sutures, interrupted single sutures, double sling sutures, or a combination of these for achieving primary closure.

### 2.3. Orthodontic Therapy

Individual treatment objectives were defined and visualized for each subject. In cases of increased tooth mobility (>grade 1), passive fixed appliances were inserted prior to periodontal therapy for stabilization. Orthodontic tooth movement was carried out using fixed orthodontic appliances and individualized segmented arch mechanics in pre-adjusted 0.022-inch slot-sized brackets. Orthodontic movement was started with a 0.012 nickel–titanium (Ni–Ti) wire, followed by the alignment with the sequence of 0.014 Ni–Ti, 0.016 Ni–Ti, 0.018 Ni–Ti and 0.016 × 0.016 stainless steel wire. Up to the sequence of 0.016, Ni–Ti wire teeth were “secured” by a figure eight ligature in order to provide continuous transmission of orthodontic forces. Maximum emphasis was put on applying low forces and moments. Bone-borne temporary anchorage devices were used in some cases for anchorage reinforcement. Once treatment goals were achieved, orthodontic appliances were removed, and teeth were stabilized with bonded fixed retainers or fiber-reinforced restorations. In all cases, target teeth were moved toward the defect.

### 2.4. Supportive Care

Frequent recall visits were scheduled at 2 days, 2 weeks and 4 weeks post-surgery. Subsequently, all patients were enrolled in a regular supportive care program every 2 months for the duration of the study. In case of recurrent periodontal inflammation, OT was interrupted until inflammation could be controlled by gentle biofilm removal and oral hygiene reinforcement.

### 2.5. Periodontal Parameters

For each center, the same expert periodontists performed RPS, expert orthodontists performed OT and the same calibrated examiners collected all clinical parameters. In addition to the previously reported data at baseline, 6 and 12 months, the following periodontal outcome variables were recorded at 24 months:(1)Clinical attachment level (CAL),(2)Probing pocket depth (PPD),(3)Bleeding on probing (BOP),(4)Full-mouth bleeding scores (FMBS),(5)Full-mouth plaque scores (FMPS).

### 2.6. Patient-Reported Outcomes

After thorough explanation of the information to be collected, perceptions of OHrQoL were assessed with a questionnaire given to all participants regarding their oral status at baseline, 6, 12 and 24 months. The twelve-question general oral health assessment index (GOHAI), originally developed by Atchinson and Dolan [42], was used as a tool of measurement in validated translations of the GOHAI questionnaire into the native language of the participants [43,44,45]. Each of the twelve questions referred to their personal experience in the previous 3 months and was answered independently by the patient using a Likert scale (0 = “never” to 4 = “very often”) (Appendix A).

For the evaluation, the answer scores for the twelve questions were summed up after coding [25], and for questions 3, 5 and 7, scoring was inverted and the scale thus ranged from 0–48 [28]. A high value indicates impairments of oral health-related quality of life, and low values indicate only a few problems.

### 2.7. Data Analysis

Computerized chairside periodontal data entry into a periodontal electronic database [Parostatus, Berlin, Germany or Florida Probe data base, USA] allowed for an export, via excel, into the statistical software program.

Descriptive statistics were summarized as means and standard deviations for quantitative data and frequencies and percentages for qualitative data. Means for each treatment group and differences between treatment groups were presented, along with associated 95% confidence intervals.

The comparison of clinical CAL changes after 24 months between treatment groups was based on a two-sided two-sample *t*-test, at the 5% level of significance. Statistical analysis of the clinical data was performed by an independent biostatistician (RF) using the software IBM^®^ SPSS^®^ Statistics 29 (Software version: 29.0.0).

## 3. Results

All 43 patients (mean age: 45.4 ± 11.9 years (early OT), 52.0 ± 9.4 years (late OT), 26 females, 17 males) were followed up until the time of their 24-month visit (until January 2022) and when they had completed all of their follow-up examinations (Figure 1). At 24 months, thirty patients had finished the combined treatment with 18 patients (78%) in the early group and 12 patients (60%) in the late group.

### 3.1. Periodontal Outcomes

Comparing the two treatment protocols, mean clinical attachment level gain (∆CAL ± SD) after 24 months was statistically significantly higher for early OT (5.96 ± 2.1; CI: 6.8, 5.1 mm) versus late OT (4.65 ± 1.76; CI: 5.4, 3.9 mm) (*p* = 0.034). When compared to 12 months, CAL showed further improvements at 24 months with an intergroup difference from the baseline of 1.31 mm in favor of early OT (Table 1).

Table 2 depicts descriptive statistics for clinical parameters at baseline (BL), 12 and 24 months. Both groups were well-balanced at baseline with regard to CAL and PPD and showed statistically significant improved outcomes after 12 and 24 months (*p* < 0.0001): Baseline CAL had changed from 9.8 to 3.9 (early OT) and from 9.2 to 4.5 mm (late OT) at 24 months, respectively. Mean probing pocket depths (PPD) remained stable between 12 and 24 months with 2.9 mm (SD: 0.9) in the early OT group and 3.2 mm (SD: 0.9) in the late OT group. Pocket closure (PPD ≤ 4 mm) was obtained in 91% of defects with early OT compared to 90% with late OT. Over the course of treatment, patients maintained their good level of adherence to a strict 2-month performed supportive care protocol and full-mouth plaque scores were consistently low (FMPS well under 20%). This was accompanied by low full-mouth bleeding scores (FMBS) of 10.5 ± 4.8% vs. 12.7 ± 6.8% (early vs. late) at baseline, 10.6 ± 4.9% vs. 7.7 ± 4.9% at 6 months, 14.7 ± 13.1% vs. 11.3 ± 9.1% at 12 months and 9.2 ± 8.2% vs. 7.2 ± 4.1% at 24 months (Table 3).

### 3.2. Oral Health-Related Quality of Life (OHrQoL)

The OHrQoL of the patients, as measured by GOHAI sum-scores, improved continuously over the course of the study from 26.1 ± 7.5 to 9.6 ± 4.7 for early OT and from 25.1 ± 7.1 to 12.7 ± 5.6 for late OT (Figure 2) without relevant differences between the two treatment groups. In a subgroup of patients that had already completed the combined perio-ortho treatment at 24 months (*n* = 18/23 for early OT and *n* = 12/20 for late OT), the final GOHAI scores showed to be very similar (8.1 ± 4.4 vs. 8.8 ± 4.7).

## 4. Discussion

The results of this 24-month follow-up of a multicenter RCT provide evidence that early stimulation of periodontal wound healing with biomechanical forces has a favorable impact on the clinical outcomes of regenerative periodontal procedures in stage IV periodontitis patients with pathologic tooth migration. The present findings also support the hypothesis that a combined periodontal regenerative and orthodontic treatment could significantly improve the oral health-related quality of life of these patients. Both of the above findings are novel and of high clinical relevance.

The scientific rationale for our previous study [10] was the limited information available on the treatment of patients with stage IV periodontitis with intrabony defects and pathologic tooth migration in need of orthodontic therapy. The optimal interval between regenerative periodontal surgery and orthodontic therapy (OT) had been a matter of ongoing debate. The principal findings after 12 months were that significant periodontal improvements of a similar magnitude were observed following early (after 4 weeks) or late (after 6 months) initiation of OT.

So far, to the best of our knowledge, no other randomized clinical study has evaluated the effect of the timing of OT on periodontal outcomes over a period of 24 months. These findings confirm a suspected “stimulating” effect of early OT on regenerative outcomes [17,18,19,23]. Both healing after RPS and healing after OT are highly coordinated processes in which various cells, such as immune, bone and periodontal ligament (PDL) cells, cytokines and signals/pathways, are involved. The whole periodontal attachment apparatus, including the alveolar bone, exhibits biological responses and changes, including a modification of the local vascularization. As known from wound healing studies, cells can respond to mechanical signals and micromechanical forces, where micro-deformations on the cellular level can stimulate cell proliferation and division [46]. In particular, PDL fibroblasts are mechano-sensing cells responsible for a complex immune response associated with the initiation of bone remodeling [47]. Fibroblasts can react to micro-deformational forces with increased proliferation and expression of collagen type I, basic fibroblast growth factor and transforming growth factor beta [48]. Mechanosensitive cells of the periodontium possess the ability to respond to a mechanical load by changing their cellular functions, including, among others, cytoskeletal rearrangement [47,49,50]. Mechanical stress of an appropriate amount which is applied to the cell membrane is detected, among others, by integrins and focal adhesion molecules, and, in this way, it triggers the assembly of specified “stress fibers” of the cytoskeleton. The latter is connected to the nuclear lamina by the linker of nucleoskeleton and cytoskeleton (LINC) complex [51]. In this way, the rearrangement of the cytoskeleton is transduced to the nucleus, leading to transcriptional changes affecting pathways that regulate cell proliferation, differentiation, motility, as well as the production of cytokines and growth factors [52].

In periodontal disease [stage III or IV], the wound associated with intrabony periodontal defects remains in the inflammatory phase and fibroblasts cannot perform their tasks due to the inflammatory environment. It is known that, in the initial phase of tooth movement, mechanical forces distort the interstitial space within the PDL and alveolar bone [53,54]. By application of micro-mechanical forces applied shortly after regenerative surgery, wound micro-deformations may induce cellular proliferation and migration [46] and enhance periodontal regeneration as well as tissue remodeling [55]. However, it has to be realized that the biological mechanisms underlying orthodontic tooth movement are still not fully understood [56]. More well-designed preclinical experiments have to be performed in order to elucidate the synergy between regenerative medicine and biomechanical force application in periodontal defects to explain the favorable clinical outcomes of the present study.

To the best of our knowledge, no other prospective study has investigated the impact of a combined periodontal–orthodontic therapy on the quality of life of patients with stage IV periodontitis affected by pathologic tooth migration. The finding of significant improvements in OHrQoL, as measured by a continuous reduction in GOHAI scores, confirms that the combined treatment not only improved the objectively assessed periodontal conditions of the patients but also their subjective perception of regained oral health, due to improved esthetics and function.

Earlier studies have already demonstrated that periodontal therapy has a positive impact on OHrQoL in patients affected by periodontitis, as measured by various accepted scoring systems [57,58,59,60]. These effects were mainly reported between 1 week and up to 12 months following the non-surgical periodontal treatment. No significant differences in the positive impact on OHrQoL were seen when comparing quadrant-wise scaling and root planing versus one-stage full-mouth disinfection [61]. Patients treated by periodontal surgery reported a worse OHrQoL in the first post-operative week [62]. With regard to the impact of periodontal surgery compared to non-surgical treatment, in some studies a low impact was observed [58,63,64,65] whereas others reported more pronounced additional improvements following surgery [66]. It has also been shown that orthodontic therapy has a positive impact on OHrQoL in patients affected by malpositioned teeth/malocclusion. Most of these studies, however, were conducted in children and adolescents [67]. Little, if any, information is available on adults [68]. Based on these reports of positive impacts on OHrQoL of orthodontic therapy in patients with malocclusion and of periodontal therapy in patients with periodontitis it can be assumed that the significant improvements observed in the present study can be attributed to both components of the combined therapy.

GOHAI sum scores, in the way they were calculated in the present study, can range between 48 (worst OHrQoL) and 0 [28,69]. The GOHAI rather than the alternative OHIP scoring system was chosen for the present study based on the findings by Öhrn and Jönsson [28]. With regard to the magnitude of the effect observed, already at 12 months after periodontal surgery mean GOHAI scores were reduced by about 6 points and decreased by another 7 (late OT) and 10 points (early OT) at 24 months, respectively. Interestingly, in the subgroup of patients in whom orthodontic treatment was completed at 24 months, the mean GOHAI scores were below 9, with no difference between the groups. In the absence of a meaningful benchmark for comparison between mean scores, Tsakos [70] proposed to employ minimally important differences (MID) instead to assist data interpretability. Following this suggestion, Jönsson and Öhrn [69] established a MID of 3 for improvements in GOHAI scores in 87 patients before, and at 12 months after, non-surgical periodontal therapy.

At present, no such MID values have been reported for the treatment of stage IV periodontitis patients. However, the significant improvements in OHrQoL (>15 GOHAI score points) in the present study compare favorably with a reduction of 5.4 points, as reported in a recent systematic review with meta-analysis, following prosthetic rehabilitation of fully/partially edentulous with previous periodontitis [71].

The present 24-month follow-up of an RCT has several strengths, such as the prospective well-controlled design, the length of follow-up, the high adherence of the patients to supportive care and compliance with a high level of self-performed oral hygiene, and the multi-center, multi-national approach, among others. On the other hand, the limited number of subjects and the fact that patients were treated in specialist settings may limit the generalizability of the results.

Future interdisciplinary studies with close cooperation between periodontics and orthodontics are warranted. A further exploration of the synergy between regenerative periodontal medicine and biomechanical orthodontic forces and their impact on the patients’ OHrQoL is of high clinical relevance because it is well-known that many adult patients affected by severe periodontitis are interested in seeking orthodontic treatment for oral rehabilitation due to the esthetic and functional changes caused by pathologic tooth migration [72].

## 5. Conclusions

Within the limitations of this study, taken together, the findings of the present 24-month follow-up show that a combined periodontal regenerative and orthodontic treatment for patients with good adherence to supportive care protocols resulted in:(1)significantly improved periodontal conditions,(2)significantly higher CAL-gain for early initiation of OT,(3)an overall significantly improved OHrQoL.

From the perspective of oral rehabilitation, orthodontic therapy plays an important role in the comprehensive treatment of stage IV periodontitis patients. More well-designed preclinical studies are warranted to further elucidate the mechanisms underlying the observed synergy between periodontal regenerative medicine and orthodontic biomechanical force application in advanced periodontitis.

## Figures and Tables

**Figure 1 bioengineering-10-00695-f001:**
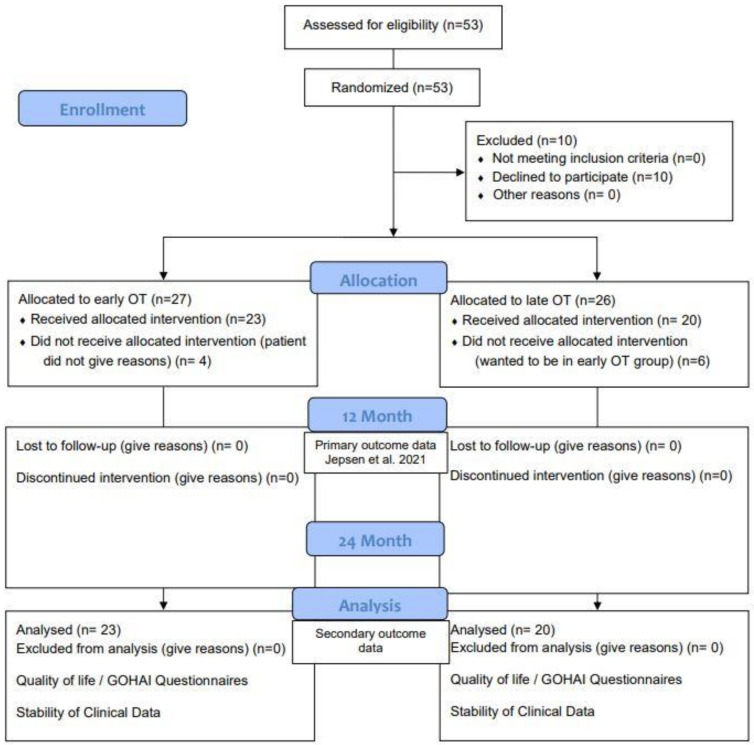
Study flowchart following CONSORT (Consolidated Standards of Reporting Trials) guidelines for clinical trials. Fifty-three patients met the inclusion criteria, 26 patients were allocated to the group with late orthodontic therapy (OT) after regenerative periodontal surgery, and 27 to the group with early OT after regenerative periodontal surgery. A total of 10 patients withdrew from the study, 6 expected to be part of the early and withdrew after allocation to the late treatment group. The other 4 patients allocated to the test group did not want to continue the study without giving any reason. All patients of the study completed their 24-month examination.

**Figure 2 bioengineering-10-00695-f002:**
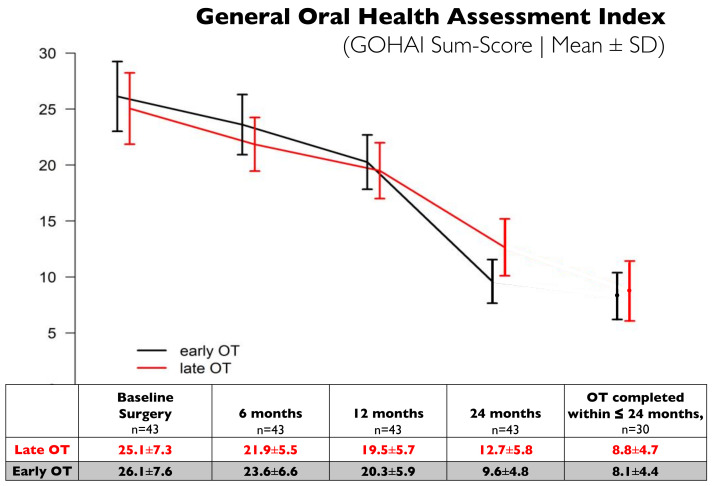
Patient-reported oral health-related quality of life (OHrQoL) as assessed by GOHAI sum-scores recorded at baseline (before surgery), at 6, 12, and 24 months. Mean GOHAI scores (± SD) in subjects (*n* = 43) treated with late OT (red line) and early OT (black line). Mean GOHAI scores (±SD) for subjects (*n* = 30) with OT completed in less or equal to 24 months are presented separately. GOHAI scores are expressed as total sum scores with all 12 questions included (values ranging from 0 to 48) with higher scores indicating greater negative impact on oral health-related quality of life [28].

**Table 1 bioengineering-10-00695-t001:** Changes in clinical parameters CAL and PPD compared to baseline at 12 and 24 months (mean ± SD) for target sites in early and late treatment group. Differences between both groups in CAL change after 24 months (secondary outcome) were tested by unpaired *t*-test.

		Early OT *n* = 23		Late OT *n* = 20		Early vs. Late OT	
		BL-12 mo	BL-24 mo	BL-12 mo	BL-24 mo	∆Change BL-24 mo	
∆CAL (mean ± SD)	mm	5.39 ± 2.2	5.96 ± 2.10	4.45 ± 1.7	4.65 ± 1.76	1.31	*p* = 0.034
*Estimate*	95% CI	[6.3, 4.4]	[6.81, 5.10]	[5.3, 3.6]	[5.42, 3.88]	[2.5, 0.1]	
∆PPD (mean ± SD)	mm	4.21 ± 1.9	4.43 ± 1.62	3.90 ± 1.5	3.90 ± 1.33	0.53	*p* = 0.248
*Estimate*	95% CI	[5.0, 3.4]	[5.1, 3.7]	[4.6, 3.2]	[4.5, 3.3]	[1.46, 0.39]	

BL: Baseline, CAL: Clinical Attachment Level, PPD: Probing Pocket Depth, CI: Confidence Interval.

**Table 2 bioengineering-10-00695-t002:** Clinical parameters (mean ± SD) for target sites in early and late orthodontic treatment (OT) group at baseline, 12 months and 24 months. Differences between follow-up visits in CAL or PPD after 24 months (secondary outcome) were tested by paired *t*-test.

Variable		Early OT (*n* = 23)				Late OT (*n* = 20)			
		Baseline	12 mo	24 mo	BL vs. 24 mo	Baseline	12 mo	24 mo	BL vs. 24 mo
CAL (mean ± SD)	mm	9.8 ± 2.5	4.4 ± 1.7	3.9 ±1.9	*p* <0.0001	9.2 ± 2.5	4.7 ± 2.4	4.50 ± 2.19	*p* < 0.0001
*Estimate*	95% CI	[8.8, 10.9]	[3.7, 5.2]	[3.1, 4.7]	[5.0, 6.7]	[8.0, 10.4]	[3.6, 5.8]	[3.5, 5.5]	[3.9, 5.5]
PPD (mean ± SD)	mm	7.3 ± 1.6	3.1 ± 0.9	2.9 ± 0.9	*p* < 0.0001	7.1 ± 1.7	3.2 ± 1.1	3.2 ± 0.9	*p* < 0.0001
*Estimate*	95% CI	[6.6, 8.0]	[2.7, 3.5]	[2.5, 3.3]	[3.7, 5.1]	[6.3, 7.9]	[2.7, 3.7]	[2.7, 3.6]	[3.1, 5.4]
PI	n (%)	4 (17%)	3 (13%)	2 (8%)		1 (5%)	2 (10%)	2 (10%)	
BOP	n (%)	13 (53%)	7 (30%)	4 (17%)		9 (45%)	3 (15%)	0 (0%)	
PUS	n (%)	1	0	0		2	0	0	
Pocket closure (PPD ≤ 4 mm)	n (%)	n/a	21 (91%)	21 (91%)		n/a	17 (85%)	18 (90%)	
Pocket closure (PPD ≤ 4 mm, no BOP)	n (%)	n/a	16 (69%)	18 (78%)		n/a	15 (75%)	15 (75%)	

BL: Baseline, CAL: Clinical Attachment Level, PPD: Probing Pocket Depth, PI: Plaque Index, BOP: Bleeding on Probing, PUS: Suppuration, CI: Confidence Interval.

**Table 3 bioengineering-10-00695-t003:** Changes of patient-based plaque (FMPS) and bleeding scores (FMBS) over time: at baseline, 6, 12 and 24 months and numbers of patients in the different phases of orthodontic therapy (1 = active, 2 = retention phase, 3 = finished).

Variables		Early OT (*n* = 23)					Late OT (*n* = 20)		
		Baseline	6 Months	12 Months	24 Months	Baseline	6 Months	12 Months	24 Months
Orthodontic therapy	Number of patients								
active	n		23	12			20	17	2
retention	n			9	5			3	6
finished	n			2	18			0	12
Full-mouth plaque scores & bleeding scores									
FMPS * (mean ± SD)	(%)	12.9 ± 4.9	15.0 ± 6	16.9 ± 10.1	13.2 ± 7.3	15.2 ± 6.2	15.0 ± 7.0	17.1 ± 8.6	13.4 ± 6.7
	95% CI	[11, 15]	[12, 18]	[13, 21]	[10, 16]	[12, 18]	[12, 18]	[13, 21]	[10, 16]
FMBS ** (mean ± SD)	(%)	10.5 ± 4.8	10.6 ± 4.9	14.7 ± 13.1	9.2 ± 8.2	12.7 ± 6.8	7.7 ± 4.9	11.0 ± 9.0	7.2 ± 4.1
	95% CI	[8, 13]	[9, 13]	[9, 21]	[5, 13]	[10, 16]	[6, 10]	[7, 15]	[5, 9]

* FMPS = full-mouth plaque score [O’Leary-1972], ** FMBS = full-mouth bleeding score (out of 6 sites per tooth).

## Data Availability

The data that support the findings of this study are available from the corresponding author upon reasonable request.

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
