# Peer review of "Synergy of Regenerative Periodontal Surgery and Orthodontics Improves Quality of Life of Patients with Stage IV Periodontitis: 24-Month Outcomes of a Multicenter RCT"

_bioengineering, 2023, doi:10.3390/bioengineering10060695_

Round 1

Reviewer 1 Report

This is a very interesting paper, well-written and well-structured with one main issue.

I have one main issue concerning a paper that was published using the same material. You need to go to this paper to understand what the authors did and how the study was set up. A QR code in the present article could be inserted in order to have access to the former article.

this article is a secondary aim of the primary study ?

In the discussion there is no mention of this first article and in the results and statistics no report of the dynamics of the first and present paper.

Concerning the surgical procedure: only the materials are described and not the procedures performed.

Concerning the orthodontic treatment: no information on the used technique, archwire sequence, etc ...

Concerning the statistics: was there a normal distribution? No details on the randomization. No description of the used statistical analysis? Mean age? Treatment time?

medical pathologies? inclusion/exclusion criteria?

There is no discussion of the intern validity.

The conclusion needs to be rewritten.

Author Response

Dear Reviewer,

Thank you for you comments, attached you'll find our our point-by-point response. Please see attachment.

BR

Karin Jepsen

Reviewer 2 Report

This is an important topic with significant clinical application and I am happy that authors recognize the need. However, a clinical trial study should be much more controlled and standardized which this article lacks. Therefore, while I admire the courage of the authors on accepting such a difficult task, unfortunately, the study is inconclusive. The main question of the article is the fact that orthodontic and periodontic treatment are not conflicting and whether the proper combination of both can be beneficial for the patient ( a major disagreement among periodontists that believe ortho treatment worsens the condition of the patient). However, it is not clear what orthodontic treatment or tooth movement is the target of the study. Does the patient receive exactly similar orthodontic treatment? did the teeth move toward the defect or away from the defect? Did movement include tilting or bodily movement, intrusion, or extrusion? How much the target teeth moved?  What was the method of calibration between all the clinicians and persons that did the measurements? what was the intra and inter-observers' difference?  were the examiners blind? what was the method of randomization? Where the clinical trial was registered? and many more.

I suggest the authors re-evaluate the article and clarify the procedures, especially the orthodontic treatment, and perhaps avoid the usage of the term, clinical trial.

Author Response

Dear Reviewer,

Thank you for you comments, attached you'll find our our point-by-point response.

BR

Karin Jepsen

Round 2

Reviewer 1 Report

Thank you for your answers and I am happy to give a positive advice